# Cleavage of LOXL1 by BMP1 and ADAMTS14 Proteases Suggests a Role for Proteolytic Processing in the Regulation of LOXL1 Function

**DOI:** 10.3390/ijms23063285

**Published:** 2022-03-18

**Authors:** Tamara Rosell-García, Sergio Rivas-Muñoz, Alain Colige, Fernando Rodriguez-Pascual

**Affiliations:** 1Centro de Biología Molecular “Severo Ochoa” Consejo Superior de Investigaciones Científicas (C.S.I.C.), Universidad Autónoma de Madrid, E-28049 Madrid, Spain; tamara.rosell@gmail.com (T.R.-G.); sergio.rivas@cbm.csic.es (S.R.-M.); 2Laboratory of Connective Tissues Biology, GIGA, University of Liège, Sart Tilman, 4000 Liège, Belgium; acolige@uliege.be

**Keywords:** extracellular matrix, elastin, lysyl oxidase-like 1, proteolysis, bone morphogenetic protein 1, a disintegrin and metalloproteinase with thrombospondin motifs

## Abstract

Members of the lysyl oxidase (LOX) family catalyze the oxidative deamination of lysine and hydroxylysine residues in collagen and elastin in the initiation step of the formation of covalent cross-links, an essential process for connective tissue maturation. Proteolysis has emerged as an important level of regulation of LOX enzymes with the cleavage of the LOX isoform by metalloproteinases of the BMP1 (bone morphogenetic protein 1) and ADAMTS (a disintegrin and metalloproteinase with thrombospondin motifs) families as a model example. Lysyl oxidase-like 1 (LOXL1), an isoform associated with pelvic organ prolapse and pseudoexfoliation (PEX) glaucoma, has also been reported to be proteolytically processed by these proteases. However, precise molecular information on these proteolytic events is not available. In this study, using genetic cellular models, along with proteomic analyses, we describe that LOXL1 is processed by BMP1 and ADAMTS14 and identify the processing sites in the LOXL1 protein sequence. Our data show that BMP1 cleaves LOXL1 in a unique location within the pro-peptide region, whereas ADAMTS14 processes LOXL1 in at least three different sites located within the pro-peptide and in the first residues of the catalytic domain. Taken together, these results suggest a complex regulation of LOXL1 function by BMP1- and ADAMTS14-mediated proteolysis where LOXL1 enzymes retaining variable fragments of N-terminal region may display different capabilities.

## 1. Introduction

The extracellular matrix (ECM) constitutes an intricate molecular network that surrounds and integrates cells and tissues in multicellular organisms. Far beyond the traditional concept of static organic material, countless studies support the notion that the ECM is a highly dynamic material that fulfills numerous functions from mechanical to signaling [1]. From a large list of extracellular matrix components, the so-called “matrisome”, collagen and elastin deserve special recognition due to their capacity to form different supramolecular complexes [2]. A critical step in the biosynthesis of collagen- and elastin-based structures is the formation of covalent associations, a post-translational modification initiated by members of the lysyl oxidase (LOX) family. LOX enzymes catalyze the oxidation of lysine and hydroxylysine residues in collagen and elastin into highly reactive aldehydes which eventually condense to form a wide variety of inter- and intra-chain cross-linkages [3]. Five different LOX enzymes have been identified in mammals (LOX, and LOX-like 1 to 4), showing a high degree of homology in the catalytic carboxy terminal end and more divergence in the rest of the sequence. Based on domain architecture and evolution, two main groups of LOX isoforms can be distinguished, LOXL2/L3/L4 superfamily, characterized by the presence of multiple scavenger receptor cysteine-rich (SRCR) domains and related to LOXL2, LOXL3 and LOXL4 isoforms; and LOX/L1/L5 superfamily, displaying pro-peptide and proline-rich regions typical of mammalian LOX and LOXL1 [4]. While a comprehensive analysis of substrate specificity for each isoform is still missing, most features observed in LOXL1 knockout mice, including pelvic organ prolapse, enlarged pulmonary alveoli, and loose skin, all manifestations of defective elastic fiber formation, suggest that LOXL1 has a prominent role in elastin cross-linking [5,6]. Genomic-wide association studies have identified LOXL1 gene as the major genetic risk factor for the development of pseudoexfoliation (PEX) glaucoma [7]. This glaucomatous degeneration arises from the accumulation of microscopic granular material that ultimately blocks normal drainage of the aqueous humor leading to increased intraocular pressure and thereby causing glaucoma. Characterization of PEX deposits have revealed the presence of elastic fiber components, such as elastin, fibrillin-1, fibulins, and LOXL1, an observation establishing a link between LOXL1 and elastin cross-linking [8,9,10]. In vitro modelling of elastogenesis has shown that LOXL1 is targeted to nascent elastic fibers through interaction with fibulin-5 where the pro-peptide region seems to support its incorporation into the microfibril scaffold [5,11,12]. In addition to this major function of LOXL1 in elastogenesis, evidence from the LOXL1 knockout mice also suggests a role of this isoform in bone collagen cross-linking where defective collagen organization and skeletal abnormalities were observed in female, but not male, mutant mice in an osteoporotic-like effect [13]. Nevertheless, while this represents valuable knowledge, there is still little information available on the regulation of LOXL1 function.

Proteolysis governs multiple processes in the synthesis and remodeling of the ECM [14]. In some cases, ECM components are multidomain proteins that are excised into different portions during maturation to fulfill various functions, and in others, pro-peptide sequences are removed to allow activation of zymogens or to release autoinhibition and thereby promote specific binding to partners. LOX isoforms are also good examples of proteolysis-regulated ECM components. Classical activation of the canonical LOX isoform involves the proteolytic removal of its pro-peptide region by procollagen C-proteinases of the BMP1 (bone morphogenetic protein 1)/Tolloid-like family, and more recent research sums up the action of N-proteinases of the ADAMTS (a disintegrin and metalloproteinase with thrombospondin motifs) family, ADAMTS2, and ADAMTS14, to the regulation of LOX [15,16,17]. Moreover, proteolytic processing by serine proteases regulates the ability of LOXL2 isoform to cross-link collagen IV [18]. Similar to LOX, its most closely related paralog, LOXL1 is claimed to be cleaved by BMP1-related proteinases [11,19,20]. However, the exact maturation site/s is/are not well defined as several cleavage sites have been reported in the literature, with no clear consensus about its precise localization. A high throughput analysis has also identified ADAMTS14 as a potential protease for LOXL1 processing [21]. Nonetheless, no further characterization of this proteolytic event was performed. 

Here, we describe that LOXL1 is proteolytically processed by BMP1 and ADAMTS14. By applying a proteomic approach, we identified the processing site/s by these proteases in the LOXL1 protein sequence. Our data show that BMP1 cleaves LOXL1 in a unique location within the pro-peptide region. In contrast, ADAMTS14 protease processes LOXL1 in at least three different sites located within the pro-peptide region and in the first residues of the catalytic domain, more C-terminal than the BMP1 site. Taken together, these results suggest a complex regulation of LOXL1 by BMP1- and ADAMTS14-mediated proteolysis and point to different capabilities of LOXL1 retaining variable fragments of the N-terminal region. We discuss here the potential contribution of these proteolytic events in the light of the available knowledge on the molecular structure and biological properties of LOXL1.

## 2. Results

### 2.1. Establishment of Cellular Models for the Expression of LOXL1 Protein

The nature of LOXL1 protein species found in cells and tissues is still a matter of confusion, with different reports describing the existence of various immunoreactive bands from 30 to 60 kDa or even heavier [11,21,22,23,24]. Here, we have analyzed the pattern of expression of LOXL1 protein from human Tenon’s capsule fibroblasts (TCF), a frequently used model for the study of LOXL1 function [21]. For these experiments, cells were left under basal conditions or stimulated with TGF-β1, a cytokine reported to induce LOXL1 expression, and cell extracts and supernatants were obtained after 1 and 4 days of incubation for Western blotting studies with a specific antibody recognizing the N-terminal portion of human LOXL1. As shown in Figure 1A, the antibody recognized two bands in cell extracts, one migrating at about 40 kDa whose expression levels did not significantly change upon induction, presumably representing a non-specific band; and other one appearing after 4 days of incubation with TGF-β1 and migrating close to 60 kDa, the theoretical molecular weight of the full-length LOXL1 protein, therefore being assigned as the intracellular form of LOXL1. The analysis of the cell supernatants showed a quite different result with a set of specific immunoreactive bands ranging from 35 to 50 kDa whose levels increased at 4 days and in the presence of TGF−β1, likely representing a complex mix of LOXL1 polypeptides and non-specific band/s. The presence of this complex array of LOXL1 species in the extracellular medium suggests that LOXL1 protein is the subject of specific post-translational modifications, likely including proteolysis by endogenous proteases. In order to establish a defined cell model for the expression of LOXL1 protein, we generated an HEK293 cell clone stably expressing a human LOXL1 construct fused to green fluorescent protein (GFP) under tetracycline-dependent control and obtained cell extracts and supernatants under basal conditions and after incubation with doxycycline, a tetracycline analog. Using a specific anti-GFP antibody, we detected in cell extracts a Dox-induced band above 75 kDa, consistent with the theoretical molecular weight of 90 kDa for the LOXL1-GFP fusion protein (Figure 1B). LOXL1 protein in supernatants appeared as a bunch of bands ranging from 40 to 75 kDa, indicating that this cell model recapitulates the complex regulation of LOXL1 expression observed in TCF. In terms of characterization of LOXL1-derived fragments, this cell clone presents a remarkable advantage as it allows the simultaneous detection of N-terminal and C-terminal fragments using anti-LOXL1 and anti-GFP antibodies as depicted in Figure 1C,D.

### 2.2. BMP1 and ADAMTS14 Proteolytically Process LOXL1

Using the HEK293-LOXL1-GFP cell clone, we applied two methodological approaches to characterize the ability of BMP1 and ADAMTS14 proteases to cleave LOXL1. On the one hand, we incubated LOXL1-GFP containing supernatants with BMP1 obtained from HEK293 cells overexpressing BMP1 and purified preparations of ADAMTS14 [16]. Figure 2A shows that BMP1 incubation resulted in a significant decrease in the levels of high-molecular-weight LOXL1 bands and the enrichment of a fragment above 50 kDa, indicating that BMP1 apparently cleaves LOXL1 in one specific site. ADAMTS14 was also able to process LOXL1 protein, giving rise to several fragments, including band/s only detected with anti-GFP antibody, which was indicative of fragment/s lacking the N-terminal portion of LOXL1. On the other hand, we co-cultured cells overexpressing LOXL1 with HEK293 clones stably expressing BMP1 and ADAMTS14. Similar to data obtained in in vitro assays, co-cultures with BMP1-expressing cells resulted in the accumulation of a fragment above 50 kDa at the expense of heavier LOXL1 species, all of them detected by the anti-LOXL1 antibody (Figure 2B). Co-culturing with ADAMTS14-expressing cells again gave several well-defined fragments including some only detected with the anti-GFP antibody (Figure 2C). So far, our results demonstrated that LOXL1 is processed by BMP1 protease presumably in one single location, whereas ADAMTS14-mediated proteolysis yielded a complex mixture of LOXL1-derived fragments.

The fusion of LOXL1 to the GFP tag provided an additional advantage as GFP-containing fragments could be efficiently immunoprecipitated with anti-GFP beads from in vitro assays with BMP1 and ADAMTS14 (Figure 3A). These fragments were then subjected to proteomic characterization to determine the cleavage sites by these proteases. For this purpose, LOXL1 protein products were trypsin-digested and peptides analyzed by mass spectrometry. As shown in Figure 3B, several LOXL1 peptides were identified that resulted from trypsin digestion at both N- and C-terminal ends (K or R in P1 position), whereas other peptides showed non-tryptic ends presumably resulting from the action of the proteases. In particular, we focused on locations where more peptides were detected after incubation with the proteases with respect to control conditions. The analysis resulted in one single processing site at 151–152 (RH/GG) for BMP1 and three distinct cleavage sites for ADAMTS14 at positions 216–217 (GA/AA), 292–293 (PD/PG), and 375–376 (PD/PN) (Figure 3B,C).

### 2.3. Expression of LOXL1 Truncated Forms for Analysis of BMP1 and ADAMTS14 Cleavage Sites

Data obtained by immunoblotting and proteomic analysis needed further evaluation. For that purpose, we designed LOXL1-GFP mutants with N-terminal deletions starting three residues from the identified cleavage sites but keeping the signal peptide to avoid disturbances in protein secretion. These LOXL1 truncated forms sequentially lose the corresponding cleavage sites as depicted in Figure 4A and a priori can be useful to confirm the proteolysis by BMP1 and ADAMTS14 enzymes. HEK293 stable cell clones were generated for every one of these constructs and LOXL1 expression tested by immunoblotting with the anti-GFP antibody in cell extracts and supernatants. As shown in Figure 4B, cell extracts revealed the presence of the corresponding intracellular LOXL1 proteins (with more or fewer signs of degradation) while the secreted forms in the supernatants were only found for the ∆36–148, ∆36–213, and ∆36–289 constructs, in addition to the full-length LOXL1. The more C-terminal truncated forms, ∆36–372 and ∆36–398, showed a severe secretion failure and were hardly detected in the extracellular medium. Then, supernatants containing secreted LOXL1 proteins were exposed to BMP1 and ADAMTS14 enzymes and the pattern of proteolytic fragments analyzed by Western blotting. As shown in Figure 5A, BMP1 cleaved the ∆36–148 form, though less efficiently than the full-length LOXL1, but it left untouched the remaining truncated forms. These data suggest that the BMP1 processing site is within the 148–213 segment, more likely at position 151–152. With respect to ADAMTS14 cleavage sites, we were able to confirm the site at 216–217, removed in ∆36–289, while the remaining sites could not be tested as ∆36–372 and ∆36–398 constructs did not express secreted forms.

## 3. Discussion

The biological action of LOX enzymes is fundamental to providing the ECM with its biomechanical properties. Low or null LOX enzymatic activity such as that observed in patients with LOX mutations or in LOX knockout mice is linked to cardiovascular abnormalities, mainly structural alterations in the arterial wall causing aortic aneurysms [5,25,26,27]. In contrast, excessive activity leading to increased matrix cross-linking is a hallmark of fibrosis, where enhanced stiffness promotes self-sustaining activation of the fibrotic process [28,29]. Therefore, the activity of LOX enzymes must be finely controlled to fulfill cellular requirements during ECM synthesis and remodeling. An important level of regulation takes place at gene expression with TGF-β1 and hypoxia as key factors to promoting increased transcription of LOX genes [30,31]. Proteolysis has also been recognized as a fundamental process for the activation of LOX isoforms. Previous reports have shown that LOXL2 isoform is proteolytically processed by serine proteases such as PACE4 to promote collagen IV cross-linking [17]. Early investigations on the canonical LOX isoform revealed that this enzyme is cleaved by the procollagen C-proteinase BMP1 to yield the mature form present in the extracellular medium [14,15]. Recent studies from our group have added the procollagen N-proteinases ADAMTS2 and ADAMTS14 to the repertoire of proteases capable of processing LOX [16]. Several sets of evidence support the notion that LOXL1 isoform is subjected to proteolytic regulation. Firstly, secreted LOXL1 protein frequently appears as a pattern of multiple bands ranging from 30 to 60 kDa [11,21,22,23,24]. Secondly, in vitro exposure of purified LOXL1 preparations to BMP1 protease was reported to yield potential proteolytic products at the expense of the initial precursor [18]. Finally, LOXL1 was identified as a candidate substrate for ADAMTS14 in a high throughput screening [20]. Here we demonstrate that BMP1 and ADAMTS14 are indeed able to proteolytically process LOXL1 and provide novel information about the location of the cleavage sites. Our data identified the BMP1 cleavage site at position 151–152 (RH/GG) in a protein segment very well conserved among LOXL1 orthologs (Figure 6). This location however contrasts with that reported in the work by Borel et al. for bovine LOXL1 where equivalent human positions at 134–135 (VG/DS) and 337–338 (SS/DT) were identified from peptide sequence analysis [18]. Available information on proteolysis by BMP1-related enzymes has shown a strong requirement for a D in P1’ position, an observation consistent with putative sites at 134–135 and 337–338 [32]. However, some bona fide substrates such as several collagen chains (collagen alpha-1 (V), alpha-1 (XI), alpha-2 (XI), and alpha-3 (V)), or the insulin-like growth factor-binding protein 3, do not fulfill this requirement showing more variability with Q, A, S, or T in the P1’ position. Without further analysis, we cannot explain the discrepancies between the study by Borel et al. and ours, to our knowledge, the first one where proteomic approaches were applied for the characterization of LOXL1 proteolysis.

With respect to ADAMTS14, multiple sequence alignment also show that the three distinct cleavage sites identified in our study at positions 216–217 (GA/AA), 292–293 (PD/PG), and 375–376 (PD/PN) are also well conserved within LOXL1 orthologs (Figure 6). A previously reported consensus sequence for ADAMTS14 cleavage has shown a remarkable presence of G, A, and P residues, which clearly resembles the sites observed in LOXL1 [20]. Strikingly, the two downstream cleavage sites, which could not be validated by truncated forms, show a high degree of homology with the ADAMTS2/TS14 site characterized by our group in the canonical LOX isoform, an observation supporting our findings [16].

LOX enzymes have been persistently reluctant to be expressed as recombinant active forms in heterologous systems and this circumstance has hampered the obtaining of crystals suitable for tridimensional structural analysis. The novel machine-learning AlphaFold approach is impacting biology as it provides strong predictions of the physical structure of proteins, including those that are very difficult to crystallize and therefore challenging for experimental determinations—for example, LOX isoforms [33]. This analysis has revealed that most of the N-terminal pro-peptide of human LOXL1 protein represents a low-confidence region (per-residue confidence metric (predicted local distance difference test (pLDDT) < 50). While this can be interpreted as low or no accuracy of the method for this particular region, it can also be a predictor of disordered residues. In fact, analysis of LOXL1 sequence with available computational systems has already suggested a prediction of disorder for most of the N-terminal region [34]. Interestingly, according to AlphaFold, the cleavage sites reported in our work either reside in this intrinsically disordered region or are solvent-exposed within the structured C-terminal domain (Figure 7), therefore presumably accessible to the protease action. Disordered is not synonymous with random regions. These flexible segments are extremely important in protein biology as they promote weak, but specific, interactions with multiple partners [35]. The fact that LOXL1 is made up of a disordered N-terminal region followed by a folded C-terminal catalytic domain suggests a mechanism by which the unstructured pro-peptide drives the interaction with specific partners and thereby brings the catalytic portion to the location/s where the enzymatic activity is needed. In fact, LOXL1 deposition onto the ECM during elastogenesis relies on the pro-peptide regions [11]. According to this scheme, proteolysis by BMP1 and ADAMTS14 can render LOXL1 species comprising the C-terminal catalytic domain and variable lengths of the pro-region that may show different interaction capabilities. For example, LOXL1 binding to fibulin-5 has been mapped to a protein segment immediately N-terminal to the catalytic domain at positions 334–402 [5]. Based on the binding capacity of this region, proteolysis by BMP1 would give a mature LOXL1 enzyme able to interact with fibulin-5, whereas full digestion with ADAMTS14, removing sequences before position 376, could seriously compromise this interaction. Actually, a similar proteolytic switch differentially regulates the binding of the canonical LOX to collagen [16].

To date, LOXL1 remains the most recognized genetic risk factor for the development of PEX glaucoma [36]. Nevertheless, despite significant progress, the molecular mechanism/s behind the pathogenic action of LOXL1 is/are still elusive. The presence of several elastic fiber components including LOXL1 in the fibrillary material forming PEX deposits suggests an alteration of the process of elastic fiber formation. It is tempting to speculate that a dysregulated proteolysis of LOXL1 yields products that may eventually promote the aggregation of elastic components and thereby the deposit of PEX material. Interestingly, a recent analysis of PEX aggregates with antibodies designed to independently detect the LOXL1 pro-region and catalytic domain showed relatively little overlap between both signals, an observation implying that most of LOXL1 protein within the PEX deposits is cleaved and that both domains positioned in separate locations [37].

In conclusion, our data provide novel knowledge about the proteolysis of LOXL1 by BMP1 and ADAMTS14 that supports a role for proteolytic processing in the regulation of LOXL1 biological function.

## 4. Materials and Methods

### 4.1. Cell Culture

Human Tenon’s capsule fibroblasts (TCF) were obtained from excised Tenon’s capsule specimens during cataract surgery. Informed consent to tissue donation was obtained and the protocol of the study was approved by the local ethics committee and adhered to the tenets of the Declaration of Helsinki for experiments involving human tissues and samples. For establishing explant cultures, the biopsies were dissected, placed in 50 mL tissue-culture flasks in Dulbecco’s modified Eagle’s medium (DMEM/Ham’s F12; Invitrogen, Darmstadt, Germany) containing 15% (vol/vol) fetal calf serum and antibiotic–antimycotic solution (Invitrogen), maintained at 37 °C in a humidified 95% air/5% CO_2_ atmosphere, and cell layers were passaged at confluence. For all experiments Tenon’s capsule fibroblasts were used at passage three to four.

Tetracycline (Tet)-inducible human embryonic kidney 293 (HEK293) cells stably expressing LOXL1-GFP, ADAMTS14 and BMP1 were generated and maintained in culture according to protocols previously published [20,38]. For co-cultures, cells from corresponding lines were trypsinized, counted, and mixed in a 50/50 proportion for seeding.

### 4.2. Construction of LOX, ADAMTS, and BMP1 Constructs and Generation of Stable Clones for Overexpression

A full-length human LOXL1 (UniProtKB-Q08397 (LOXL1_HUMAN)) construct in pcDNA3 vector was obtained from Wolfgang Jarolimek and Alberto Buson (Pharmaxis Ltd., Australia). The LOXL1 insert from this vector was inserted into pEGFP-N1 (Life Technologies) and the LOXL1-GFP chimera was transferred to pcDNA5/FRT/TO for expression in the Flp-In T-REx 293 system. For the generation of a stable cell line overexpressing LOXL1-GFP, the construct was co-transfected with the Flp recombinase expression plasmid pOG44 into the Flp-In T-REx 293 cell line using Lipofectamine 2000 (Invitrogen). These cells stably express the Tet repressor and contain a single integrated FRT (Flp recombination target) site. Flp recombinase expression from the pOG44 vector mediates insertion of the cDNA cassette into the genome at the integrated FRT site through site-specific DNA recombination. After 48 h, cells were selected for hygromycin B resistance (Roche Diagnostics, Barcelona, Spain), and clones appeared after 10–15 days. Isogenic pooled clones were expanded and checked for transgene expression after 48 h of incubation in the absence or presence of doxycycline (tetracycline analog) at 1 μg/mL. Specific deletions in full-length LOXL1-GFP to generate truncated forms in LOXL1 protein segments were done by site-directed mutagenesis using methods previously described [39]. All constructs were verified by sequencing.

BMP1- and ADAMTS14-overexpressing HEK293 cell clones were generated as described previously [16,20].

### 4.3. Protein Analysis

For protein studies, TCF were seeded on 100 mm culture dishes and incubated with 5 mL of serum- and phenol red-free medium containing with 5 ng/mL of TGF-β1 for 1 and 4 days. To induce transgene expression in HEK293, cells were incubated with serum- and phenol-red free medium containing 1 μg/mL doxycycline (Dox) for 48 h. Cell supernatants (5 mL) were collected and concentrated down to about 200 μL using Amicon Ultracentrifugal filters (Ultracel-10K, EDM Millipore). Separately, cell monolayers were washed with phosphate buffered saline (PBS) and lysed with 500 µL Tris-SDS buffer (60 mM Tris-HCl, pH 6.8, 2% sodium dodecyl sulphate (SDS)) to obtain cell extracts. Protein concentration for cell lysates was determined by the BCA^TM^ Protein Assay Kit (Thermo Scientific, Rockford, IL). For analysis of LOXL1 and/or GFP expression, proteins (30 μg protein for lysates or 40–50 μL for supernatants) were fractioned on SDS-polyacrylamide gels using a Mini Protean electrophoresis cell system (Bio-Rad). Gels were then transferred onto nitrocellulose membranes at 12 V for 20 min in a semi-dry Trans-Blot Turbo system (Bio-Rad). Membranes were blocked by incubation for 30 min with 1% bovine serum albumin (BSA) in Tris-buffered saline (TBS) containing 0.5% Tween-20, and antigens were detected using specific primary antibodies against LOXL1 (rabbit polyclonal antibody against LOXL1 kindly provided by Ursula Schlötzer-Schrehardt, University of Erlangen, Germany), anti-GFP (Cat.No. 11814460001, mouse monoclonal, Roche) and β-actin for normalization purposes (Sigma, mouse monoclonal) [40,41]. Blots were then developed using corresponding IRDye 680- or Li-Cor IRDye 800-labeled secondary antibodies with the Odyssey infrared imaging system (Li-Cor).

LOXL1-GFP protein products were immunoprecipitated using GFP-Trap agarose beads (ChromoTek, Planegg-Martinsried, Germany) following manufacturer’s protocol. Immunoprecipitated proteins were analysed by Western blotting with anti-GFP antibody and used further for proteomic studies.

### 4.4. Proteomic Analysis

For the analysis of the proteolytic processing sites of LOXL1 by BMP1 and ADAMTS14 enzymes, LOXL1-GFP supernatants were exposed to the proteases for 2 h at 37 °C and the reaction was then applied onto 1.2 cm wide wells of a conventional SDS-PAGE gel (0.75 mm thick, 4% stacking, and 10% resolving). The run was stopped as soon as the front entered 3 mm into the resolving gel, so that the polypeptides became concentrated in the stacking/resolving gel interface. The unseparated protein bands were visualized by Coomassie staining, excised, cut into cubes (2 × 2 mm), and placed in 0.5 mL microcentrifuge tubes. The gel pieces were destained in acetonitrile:water (ACN:H_2_O, 1:1), incubated with 10 mM DTT for 1 h at 56 °C for disulfide bond reduction and with 50 mM iodoacetamide for 1 h at room temperature in darkness for thiol group alkylation, and then digested in situ with sequencing grade trypsin (Promega, Madison, WI) as previously described [40]. Digestion products were desalted onto OMIX Pipette tips C18 (Agilent Technologies) and analyzed by RP-LC-MS/MS in an Easy-nLC II system coupled to an ion trap LTQ-Orbitrap-Velos-Pro hybrid mass spectrometer (Thermo Scientific). Peptide identification from raw data was carried out using PEAKS Studio 6 software (Bioinformatics Solutions Inc.) [41]. Database search was performed against UniProt-Homo Sapiens.fasta (decoy-fusion database). The following constraints were used for the searches: tryptic cleavage after Arg and Lys, up to two missed cleavage sites, allows non-specific cleavage at one end of the peptide, and tolerances of 20 ppm for precursor ions and 0.6 Da for MS/MS fragment ions. The variable modifications allowed were methionine oxidation and cysteine carbamidomethylation. False discovery rate was set as <1% (FDR < 0.01). Detected LOXL1 sequences were screened for non-tryptic internal peptides (no K or R in P1) as potential sites for proteolytic processing.

## Figures and Tables

**Figure 1 ijms-23-03285-f001:**
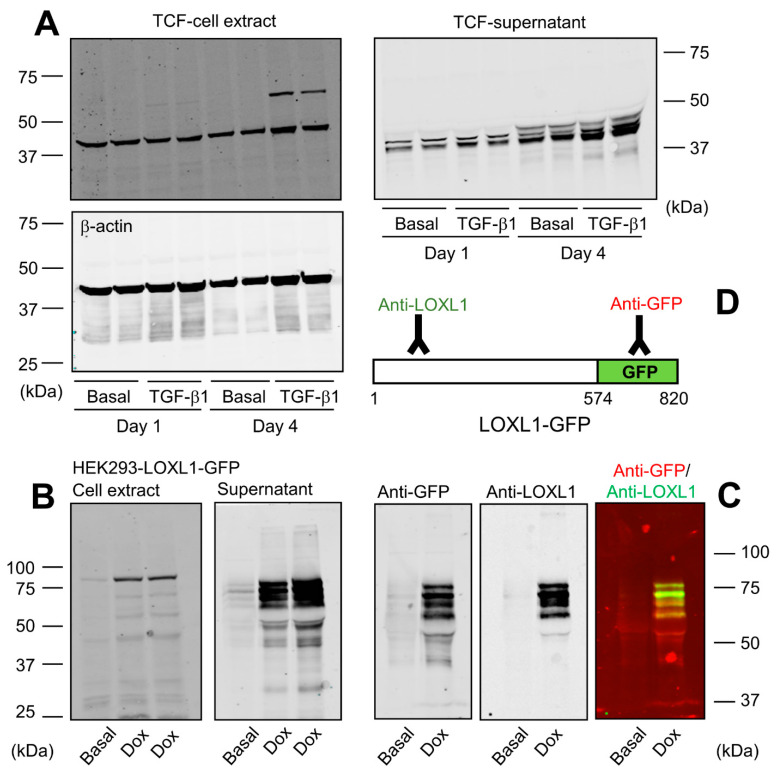
LOXL1 protein is expressed in multiple forms. (**A**) Human Tenon’s capsule fibroblasts (TCF) were incubated for 1 or 4 days under basal conditions or in the presence of 5 ng/mL of TGF-β1. Cell extracts and supernatants were taken and processed for Western blotting with a specific LOXL1 antibody recognizing the N-terminal region. Housekeeping β-actin protein was also probed in cell extracts as indicated for normalization purposes. (**B**) Generation of HEK293 cells over-expressing human LOXL1-GFP fusion protein in a tetracycline-inducible manner. LOXL1 protein in the cell extract and supernatant fractions was analyzed under basal conditions and upon a 48-h induction with the tetracycline analogue, doxycycline (Dox), using an anti-GFP antibody. Analysis of cell extracts revealed intracellular bands close to the theoretical molecular weight of 60 kDa for LOXL1 from TCF and 90 kDa for LOXL1-GFP in HEK293 cells. A band of about 40 kDa recognized in TCF extracts by the LOXL1 antibody presumably represents a non-specific product. The supernatant of TCF and HEK293 cells revealed the presence of multiple forms of LOXL1 (35–50 kDa) and LOXL1-GFP (40–75 kDa), respectively, with TCF likely also showing non-specific band/s. (**C**) Simultaneous detection of LOXL1-GFP protein with anti-GFP and anti-LOXL1 antibodies showing both signals separately and together (red, GFP; green, LOXL1). (**D**) Schematic representation of the recognition regions of LOXL1-GFP by anti-LOXL1 and anti-GFP antibodies. The blots shown correspond to representative experiments performed twice with two independent preparations.

**Figure 2 ijms-23-03285-f002:**
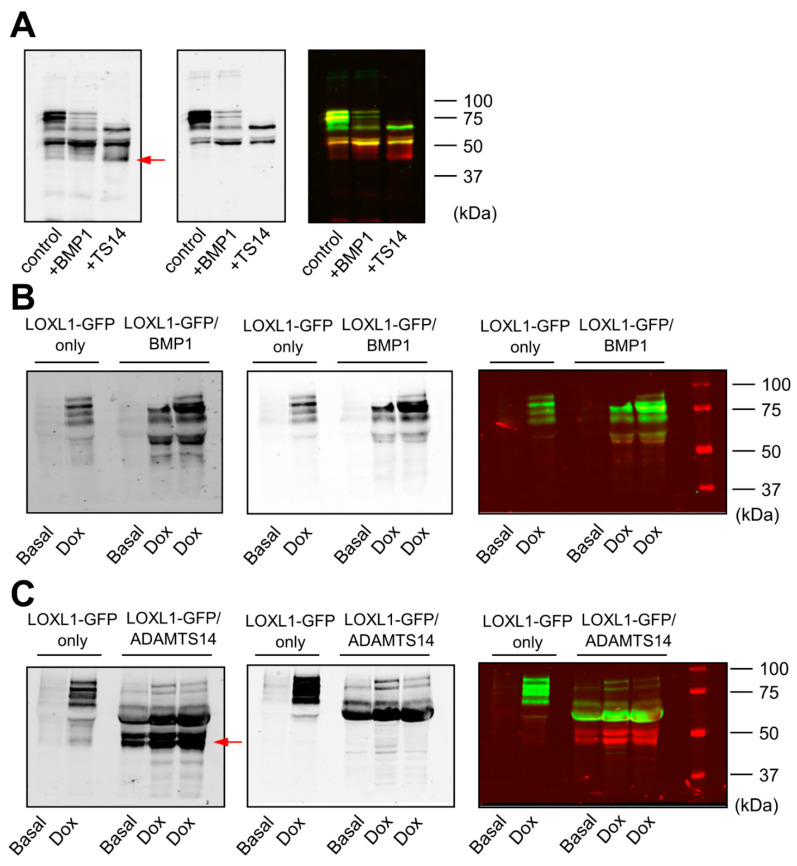
Cleavage of LOXL1-GFP by BMP1 and ADAMTS14. LOXL1-GFP cleavage was assayed by Western blotting in in vitro incubation assays (**A**) and co-cultures of LOXL1-GFP/BMP1 (**B**), or LOXL1-GFP/ADAMTS14-overexpressing cells (**C**). For in vitro assays, LOXL1-GFP-containing supernatants were incubated with BMP1 or ADAMTS14 for 2 h at 37 °C. Co-culture experiments were performed by seeding cells in a 50/50 proportion. Upon confluency, cells were left under basal conditions or stimulated with doxycycline for 48 h. Controls for single LOXL1-GFP cultures were also included (LOXL1-GFP only). In vitro reactions or concentrated supernatants from co-cultures were analyzed by Western blotting using anti-LOXL1 and anti-GFP antibodies. Blots showed detection signals for anti-GFP (left panel), anti-LOXL1 (middle panel), and both antibodies together (right panel). Note that proteolysis with ADAMTS14 yielded bands that were only recognized by the anti-GFP antibody (marked by a red arrow in anti-GFP or visualized as red bands in anti-GFP/anti-LOXL1) representing proteolytic products lacking the N-terminal portion of LOXL1. The blots shown correspond to representative experiments performed twice with two independent preparations.

**Figure 3 ijms-23-03285-f003:**
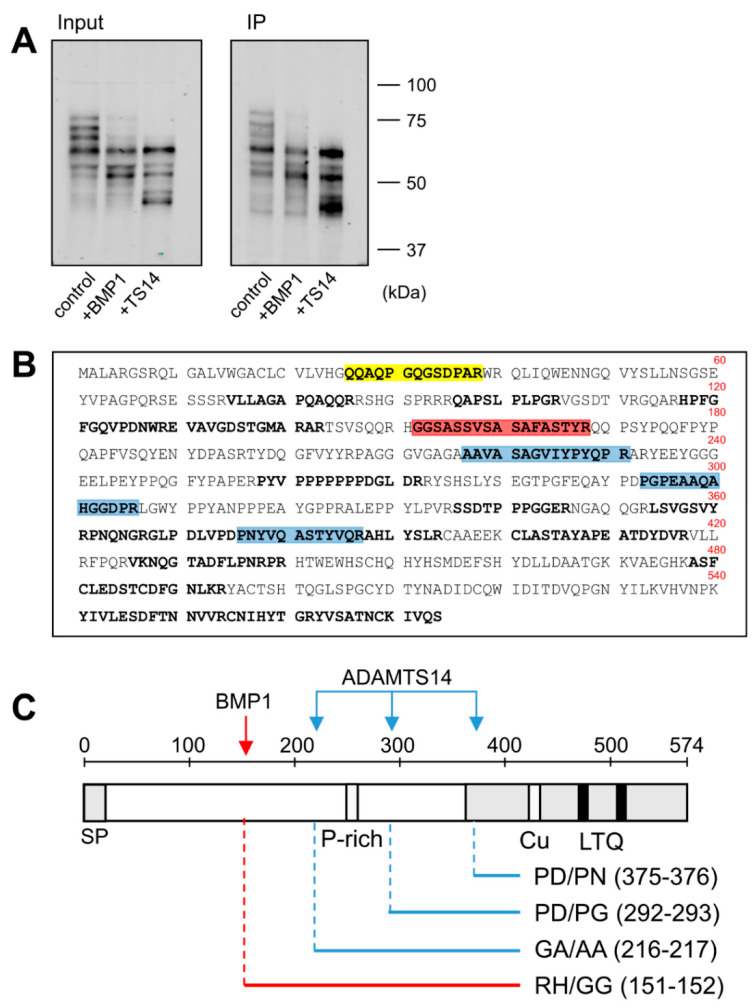
Proteomic characterization of the BMP1- and ADAMTS14-mediated cleavage sites in LOXL1 protein. LOXL1-GFP supernatants were exposed in vitro to BMP1 and ADAMTS14 for 2 h at 37 °C and proteolytic products immunoprecipitated with GFP-Trap agarose beads. Aliquots from input and immunoprecipitated samples (IP) were analyzed by Western blotting with the anti-GFP antibody (**A**). The blots shown correspond to representative experiments performed twice with two independent preparations. IP products were then trypsin-digested, and the resulting peptides fractionated by liquid chromatography and analyzed by mass spectrometry. (**B**) Peptide coverage along the human LOXL1 sequence showing identified peptides (shown in bold). In addition to tryptic fragments resulting from the action of trypsin at both ends (K or R in P1), other peptides showed non-tryptic ends and resulted by the action of other proteases. Peptide marked in red showed a non-tryptic N-terminal end was more abundantly detected under BMP1 digestion, presumably representing the cleavage site by the BMP1 enzyme. Three peptides labeled in blue also showed non-tryptic N-terminal ends and were found upon digestion with ADAMTS14. Peptide in yellow with a non-tryptic N-terminal end corresponds to the mature LOXL1 protein after removal of the signal peptide. (**C**) Schematical representation of the cleavage sites by BMP1 (red arrow) and ADAMTS14 (blue arrows) within LOXL1 sequence with information about their positions and the residues involved in the proteolysis.

**Figure 4 ijms-23-03285-f004:**
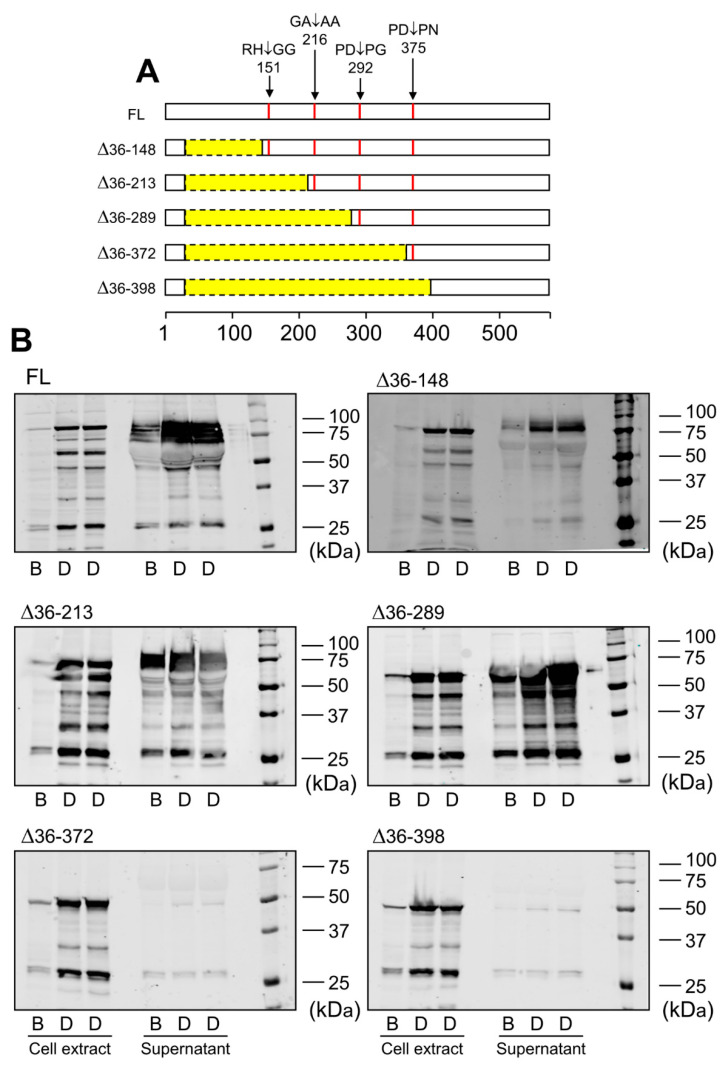
Expression of LOXL1 truncated forms of LOXL1-GFP protein. (**A**) Schematical representation of truncated forms of LOXL1-GFP protein within the LOXL1 sequence (deletions marked in yellow) showing the position of the cleavage sites in red. The set of constructs include decreasing portions of the N-terminal end of LOXL1 protein but keep the signal peptide for proper transit through Golgi and secretion. (**B**) Western blotting analysis of the expression of the full-length (FL) and indicated truncated forms of LOXL1 protein as assessed with the anti-GFP antibody under basal conditions (B) and upon stimulation with doxycycline (D) in cell extracts and supernatants. Note that all of the constructs gave rise to the corresponding intracellular LOXL1-GFP form as detected in the cell-extract fractions. However, analysis in the cell supernatants revealed that the more C-terminal truncated forms, ∆36–372 and ∆36–398 failed to appear as extracellular secreted proteins. The blots shown correspond to representative experiments performed twice with two independent preparations.

**Figure 5 ijms-23-03285-f005:**
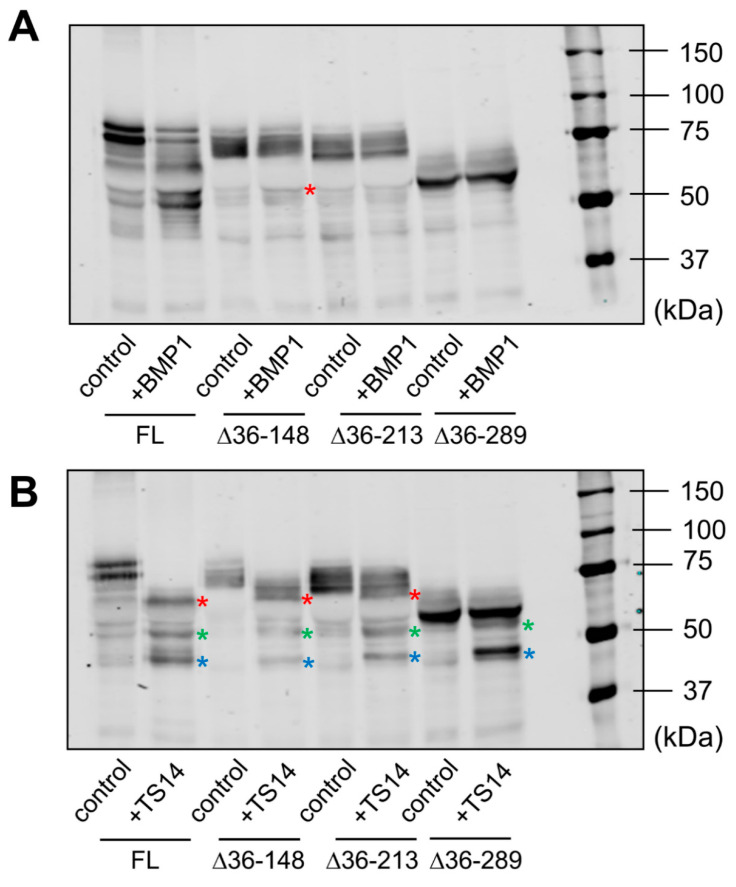
Analysis of BMP1- and ADAMTS14-mediated proteolysis in LOXL1 truncated forms. Secreted forms of full-length (FL) and truncated LOXL1-GFP proteins were exposed to BMP1 (**A**) or ADAMTS14 (**B**) for 2 h at 37 °C and analyzed by Western blotting using the anti-GFP antibody. Note that BMP1 cleaved the ∆36–148 form, though less efficiently compared to the full-length form (red asterisk in panel A indicates the proteolytic product). The remaining forms were apparently not cleaved by BMP1. Digestion with ADAMTS14 revealed a set of proteolytic products labelled with red, green, and blue asterisks in panel B. Lack of red-labeled product in the ∆36–289 construct confirmed the ADAMTS14 cleavage site at the 216–217 position. The blots shown correspond to representative experiments performed twice with two independent preparations.

**Figure 6 ijms-23-03285-f006:**
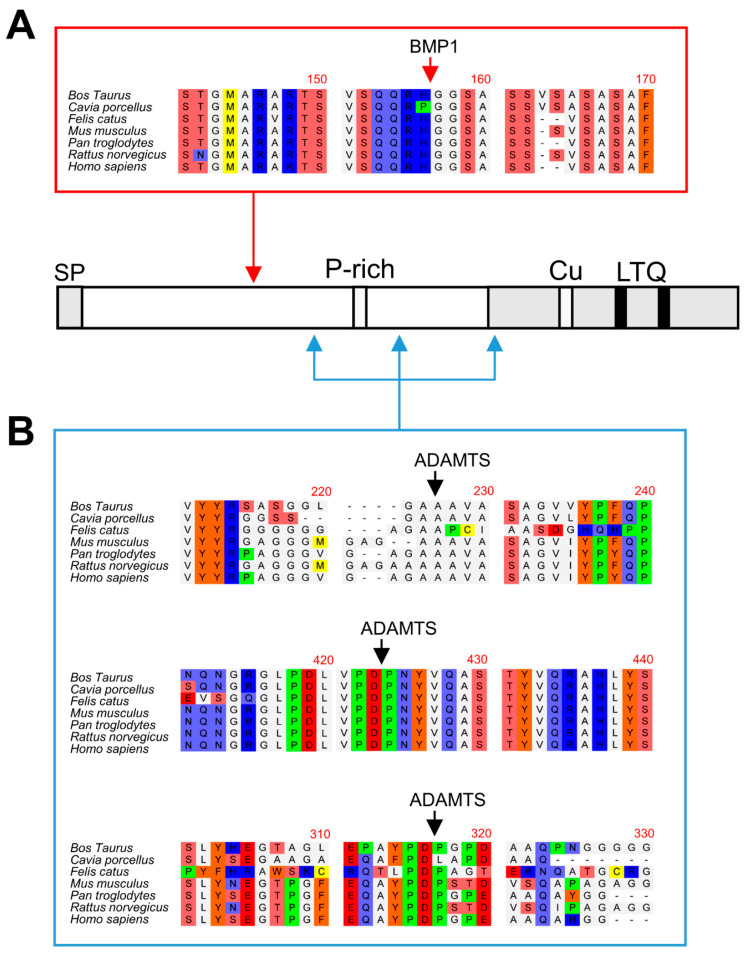
Multiple sequence alignments of BMP1 and ADAMTS14 cleavage sites. Sequence alignment of LOXL1 orthologs from the indicated species showing high homology at the BMP1 (**A**) and ADAMTS14 (**B**) cleavage sites. Protein alignments were done with the ClustalW2 program (http://www.ebi.ac.uk).

**Figure 7 ijms-23-03285-f007:**
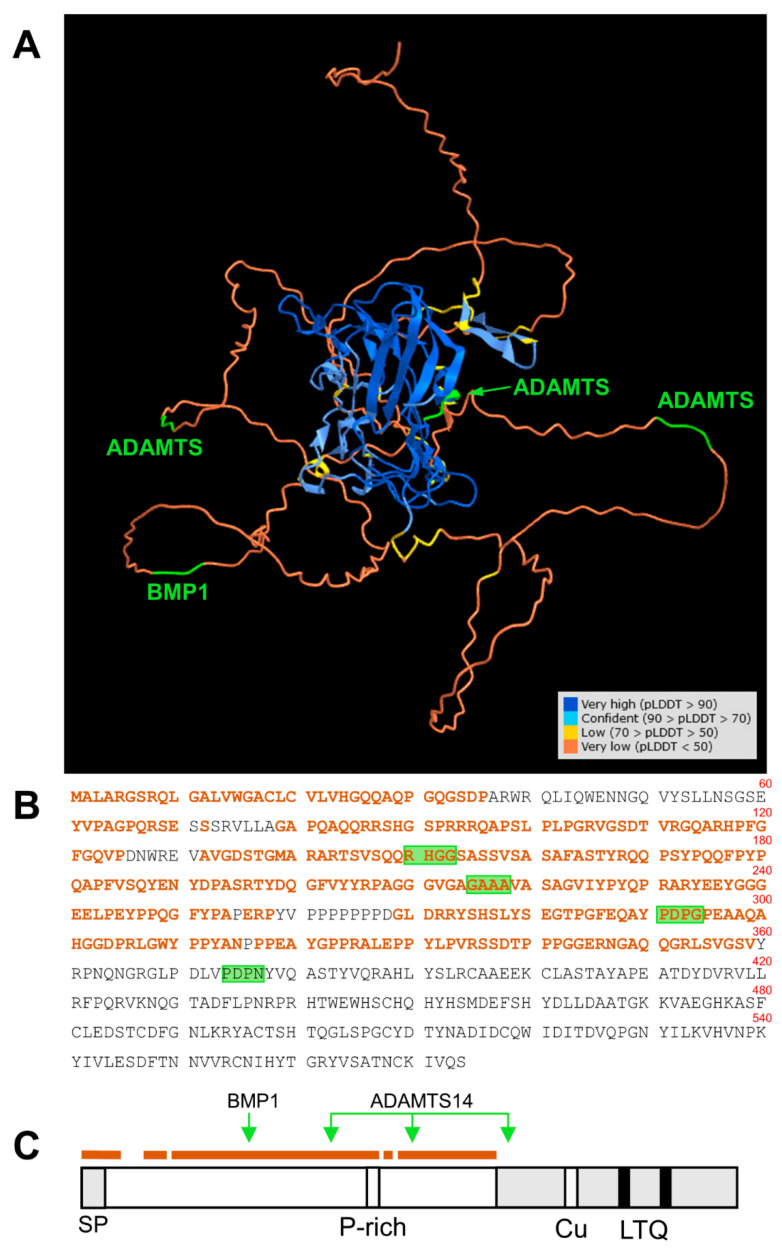
Location of BMP1 and ADAMTS14 cleavage sites within the tridimensional structure of LOXL1 protein as predicted by AlphaFold. (**A**) AlphaFold prediction model for LOXL1 protein showing the degree of confidence (predicted local distance difference test, pLDDT) with a color code as indicated (https://alphafold.ebi.ac.uk/). The N-terminal region up to position 360 mostly displays an unstructured folding (pLDDT < 50, brown), a predictor of a disordered region. This unstructured segment is highlighted within the LOXL1 protein sequence (**B**,**C**). The position of the BMP1 and ADAMTS14 cleavage sites is indicated in green. Note that these processing sites are within the disordered region (BMP1 at 151–152, ADAMTS14 at 216–217, and 292–293), or solvent-exposed within the structure C-terminal domain (ADAMTS14 at 375–376).

## Data Availability

Data available on request from the authors.

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
