# Peer review of "Cleavage of LOXL1 by BMP1 and ADAMTS14 Proteases Suggests a Role for Proteolytic Processing in the Regulation of LOXL1 Function"

_ijms, 2022, doi:10.3390/ijms23063285_

Round 1

Reviewer 1 Report

The study explores the proteolysis of LOX1 by BMP1 and ADAMTS14 which lends insight into development of PEX glaucoma via LOX1. This is a thorough and well described study.  

Author Response

Reviewer #1: The study explores the proteolysis of LOX1 by BMP1 and ADAMTS14 which lends insight into development of PEX glaucoma via LOX1. This is a thorough and well described study.

Thank you for your nice comments.

Final note:

We appreciate that the reviewer called our attention to the English language and style. We have extensively revised and corrected the manuscript for the revised version.

Reviewer 2 Report

The manuscript by Rosell-García and co-workers addresses an unresolved issue regarding LOXL1 biochemistry. Through an elegant approach involving proteomic analysis the authors have clarified the proteolytic processing affecting LOXL1. The manuscript is clear and well-written and only minor questions should be answered:

  1. The authors detected the presence of a probably non-specific band in western assays using cell extracts. Does this band dissapear in cells silenced for LOXL1?
  2. The antibody against LOXL1 is non-commercial. It has been previously validated? Why the authors did not use a commercial antibody ? Please provide a reference of previous studies using this antibody
  3. Western blot analyses do not include a control for equal loading such as Pounceau staining or a housekeeping protein such as GAPDH, Tubulin or similar. Please include it at least in some figures such as those from Figure 1A.
  4. How western blots with cell supernatants were performed? Were they adjusted by the protein content of the supernatant or by the number of cells/protein concentration of cell monolayers?
  5. Legend for Figure 4: Please define the terms "B" and "D" in Western blots.
  6. Discussion. Could ADAMTS2 process LOXL1 similarly to ADAMTS14?
  7. Because the proteolytic processing of LOXL1 by ADAMTS14 affects the catalytic domain, could LOXL1 activity be reduced by this mechanism?

Reviewer 3 Report

Members of the LOX family catalyse the formation of covalent cross-links in collagen and elastin which is an essential process for the maturation of connective tissue. Proteolysis serves as a regulation of LOX enzymes. Precise Molecular information about these proteolytic events is missing. Authors' data suggest a complex regulation of LOXL1 function by BMP1- and ADAMTS14-mediated proteolysis where LOXL1 enzymes retaining variable fragments of the N-terminal region may display different capabilities. This manuscript reviews 41 articles regarding this topic. The topic of this manuscript is up to date, interesting and well suited for your journal. The manuscript is well written and divided into 4 parts, the text is clear and easy to read. For better visualisation authors used 7 figures. I appreciate added photographic material which enriches this manuscript. I suggest checking for some spelling mistakes and grammar errors. Otherwise, I have no major concerns about this manuscript and I recommend it for publication.

Author Response

Reviewer #3: Members of the LOX family catalyze the formation of covalent cross-links in collagen and elastin which is an essential process for the maturation of connective tissue. Proteolysis serves as a regulation of LOX enzymes. Precise molecular information about these proteolytic events is missing. Authors' data suggest a complex regulation of LOXL1 function by BMP1- and ADAMTS14-mediated proteolysis where LOXL1 enzymes retaining variable fragments of the N-terminal region may display different capabilities. This manuscript reviews 41 articles regarding this topic. The topic of this manuscript is up to date, interesting and well suited for your journal. The manuscript is well written and divided into 4 parts, the text is clear and easy to read. For better visualization authors used 7 figures. I appreciate added photographic material which enriches this manuscript. I suggest checking for some spelling mistakes and grammar errors. Otherwise, I have no major concerns about this manuscript and I recommend it for publication.

Thank you for your nice comments.

Final note:

We appreciate that the reviewer called our attention to the English language and style. We have extensively revised and corrected the manuscript for the revised version.